# OpenReview forum: "Rethinking Visual Autoregressive Sampling with Information-Grounding Guidance"
_ICML.cc/2026/Conference — ICML 2026 regular_

### Official Review · Reviewer_Yhp1 · 2026-03-12

**Soundness:** 3
**Presentation:** 3
**Significance:** 3
**Originality:** 4
**Overall Recommendation:** 4
**Confidence:** 4

**Summary:**

This paper proposes Information-Grounding Guidance (IGG) as an alternative to CFG for scale-wise autoregressive models. The authors observe that classifier-free guidance in scale-wise autoregressive models is often dispersed and misaligned. The authors argue that not all tokens should be guided equally, and introduce a novel guidance machanism that anchor guidance signals to semantically important tokens. Extensive experiments on class-conditioned generation and text-to-image generation demonstrate that IGG consistently outperforms baseline CFG, while adding negligible computational overhead.

**Compliance With Llm Reviewing Policy:**

Affirmed.

**Final Justification:**

I do not have any other concerns, so I maintain the positive score.

**Key Questions For Authors:**

See weaknesses.

**Limitations:**

No.

**Strengths And Weaknesses:**

**Strengths**:

1. Novelty. The idea that not all tokens should be guided equally is insightful. It makes sense to allocate stronger guidance to semantically important regions.

2. Strong Empirical Results. IGG consistently outperforms CFG across class-conditioned generation and text-to-image generation tasks.

3. Computational Efficiency. The computational overhead introduced by IGG is negligible.

**Weaknesses**:

1. Missing rationale for weighting function f_k. The design of the weighting function (in Equation 6) is highly heuristic. Providing more theoretical analysis on the properties of this specific design, or including an ablation study comparing it to alternative weighting formulations, would strengthen the paper and justify the design choice.

2. Inconsistency in the formulation of the guidance schedule. The guidance scale parameter is absent in Equation 5. Yet, in Section 6.4, there is an extensive ablation study on different guidance schedules. It is confusing where the related parameters fit into Equation 5.

3. Possible dimensionality mismatch in Equation 6. The softmax operation should yield a dense attention matrix of size [Number of tokens, Number of tokens]. However, f_k is defined as outputting a single spatial weight per token. Is there a missing reduction step to collapse it down to [Number of tokens]?

---

> ### Author Rebuttal · Authors · 2026-03-31
>
> We thank the reviewer for their encouraging comments. Regarding their feedback:
>
> # [W1] Rationale for weighting function $f_k$
>
> We argue that the choice of attention-based mechanism is well-motivated. In particular, there are three reasons:
> - **Limited options to consider:** an effective implementation needs the ability to distinguish important tokens not only on pixel-level but also the intermediate latent scales in the middle of the sampling process of SwAR, where the underlying visual semantics are considerably noisier. To the best of our knowledge, there is currently no alternative method designed for this specific purpose.
> - **Well-established:** Given the standard usage of the attention mechanism to determine the relative importance of tokens at latent layers (of Transformer-based architectures), we concluded that it would be reasonable to adapt this scheme into IGG.
> - **Efficiency:**  Since we are guiding the model at inference time, efficiency is another highly important factor. Given that the computation of attention is already highly optimised in modern hardware and through specialised algorithms (e.g., FlashAttention), adopting this mechanism for IGG ensures that our method has negligible impact on sampling runtime.
>
> We did try with other approaches such as the box sampling weighting scheme, where we computed the distributions of the bounding boxes within the ImageNet dataset and sampled these boxes as targets for applying guidance (tokens falling outside these boxes are ignored). However, the performance worsened compared to vanilla CFG; the reason is that it distributes the guidance in rectangular and square shapes which does not help to sharpen intricate object details. In another approach, we tried to use a random segmentation map for weighting, yet this scheme also did not work as well since it fails to target the foreground whenever the model does not generate the same object as the segmentation map. Overall, attention-based mechanism is the best in helping to focus guidance on important details while also adapting to changes in object shapes.
>
> # [W2] Inconsistency in the formulation of the guidance schedule
> We thank the reviewer for pointing this out. The correct formulation of Equation 5 indeed should include the guidance scale parameter $\gamma$ (along with the adaptive weights $f_k$). We will add this parameter when revising the manuscript.
>
> # [W3] Dimensionality concern in Equation 6
> We affirm that there is no dimensionality mismatch. Let $H \times W$ be the number of tokens at some arbitrary scale (generation step) and $V$ be the token vocabulary size. In practice, the guidance signals at each scale are stored as a tensor of shape $[H, W, V]$. Then, $f_k$ computes the dot-product attention scores amongst the tokens, resulting in a tensor of shape $[H, W, H, W]$. Finally, these scores are used to scale the original guidance signals, resulting in a tensor of shape $[H, W, V]$. Specifically, we perform the following tensor contractions:
>
> - Dot-product attention: $[H, W, V] \times [V, H, W] = [H, W, H, W]$
> - Guidance signal scaling: $[H, W, H, W] \times [H, W, V] = [H, W, V]$
>
> Both operations can be implemented using `einsum` in PyTorch.

---

> > ### Author Rebuttal · Reviewer_Yhp1 · 2026-04-03
> >
> > Thanks to the authors' great efforts to address my concerns. Currently, I do not have any more questions. Since I have given a positive score originally, I decided to maintain the score.

---

### Official Review · Reviewer_Gw64 · 2026-03-12

**Soundness:** 3
**Presentation:** 2
**Significance:** 3
**Originality:** 4
**Overall Recommendation:** 4
**Confidence:** 3

**Summary:**

This paper addresses the misaligned and dispersed guidance signals in Scale-wise Autoregressive (SwAR) models by proposing Information-Grounding Guidance (IGG). Unlike traditional Classifier-Free Guidance (CFG), which treats all tokens equally, IGG employs a self-attention mechanism to dynamically amplify guidance signals specifically for semantically important regions. Extensive experiments demonstrate that IGG significantly outperforms CFG in both class-conditioned and text-to-image generation, yielding higher fidelity, better semantic alignment, and improved handling of complex prompts with negligible computational overhead.

**Compliance With Llm Reviewing Policy:**

Affirmed.

**Key Questions For Authors:**

Please refer to the weaknesses.

**Limitations:**

yes

**Strengths And Weaknesses:**

Strengths:
IGG effectively rectifies the inherent weakness of SwAR models where guidance dilutes across scales, resulting in superior image quality and prompt adherence compared to standard CFG and even competitive diffusion baselines; its ability to focus on foreground features enhances structural coherence without imposing significant inference costs, making it a highly practical and efficient upgrade for existing autoregressive frameworks.

Weaknesses:
1. Grounded in the motivation that "not all tokens should be equally guided," the paper exclusively relies on an attention-based mechanism to realize IGG, without exploring alternative implementation schemes that could achieve similar token-wise differentiation. Furthermore, the study lacks a comprehensive experimental investigation into guidance strategies across different generation steps (scales), leaving the optimal temporal dynamics of the guidance signal under-explored.
2. The backbone model employed in the experiments lags significantly behind current state-of-the-art (SOTA) open-weight image generation models. Consequently, the adaptability of the proposed method when deployed on more powerful, modern architectures remain questionable.
3. Although the ablation studies provide meaningful empirical comparisons, they fall short in offering a deep theoretical analysis or mechanistic explanation for the observed results, leaving the underlying causes of the method's behavior largely unexplored.

---

> ### Author Rebuttal · Authors · 2026-03-31
>
> We thank the reviewer for their encouraging comments. Regarding their feedback:
>
> # [W1.1] Reasons for choosing attention-based implementation
>
> We argue that the choice of attention-based mechanism is well-motivated. In particular, there are three reasons:
> - **Limited options to consider:** an effective implementation needs the ability to distinguish important tokens not only on pixel-level but also the intermediate latent scales in the middle of the sampling process of SwAR, where the underlying visual semantics are considerably noisier. To the best of our knowledge, there is currently no alternative method designed for this specific purpose.
> - **Well-established:** Given the standard usage of the attention mechanism to determine the relative importance of tokens at latent layers (of Transformer-based architectures), we concluded that it would be reasonable to adapt this scheme into IGG.
> - **Efficiency:**  Since we are guiding the model at inference time, efficiency is another highly important factor. Given that the computation of attention is already highly optimised in modern hardware and through specialised algorithms (e.g., FlashAttention), adopting this mechanism for IGG ensures that our method has negligible impact on sampling runtime.
>
> # [W1.2] Investigation of guidance strategies
>
> We affirm that we have explored alternative guidance strategies in the paper. In addition to the default ratio-based guidance schedule adopted from VAR (where guidance strength grows at each scale, according to the source code provided by the authors [1]), we also evaluated IGG using a fixed guidance schedule (where guidance strength is constant across all scales) and a mixed guidance scheme (where CFG and IGG are integrated). The results below (taken from Section 6.4) show that the default strategy remains the most optimal choice.
>
> | Strategy | FID |
> | - | - |
> | Vanilla (ratio-based schedule, single scheme) | **1.92** |
> | Fixed schedule | 4.29 |
> | Mixed scheme | 5.48 |
>
> Originally, we also attempted to adapt the Interval Guidance scheme for diffusion models [2]. However, as this change degraded model performance, we omitted it from the manuscript.
>
> [1] https://github.com/FoundationVision/VAR/blob/main/models/var.py
>
> [2] Kynkäänniemi, T., Aittala, M., Karras, T., Laine, S., Aila, T., & Lehtinen, J. (2024). Applying guidance in a limited interval improves sample and distribution quality in diffusion models. Advances in Neural Information Processing Systems, 37, 122458-122483.
>
> # [W2] Concern regarding adaptability of IGG
>
> While it is true that AR models (e.g., VAR, Switti) are currently not the SOTA in open-source image/video generation (flow-matching models have since beat them), it does not mean that the research problems on these models are not useful. From our knowledge, there are a number of closed-sourced models which are comparably competitive with open-source counterparts that are based on AR modelling, such as GPT-4o. Our research helps to provide insights into AR-generative models in general. Furthermore, as a researcher, we can not always follow SOTA which might lead to “local optima” of the research direction. Instead, we are trying to detect and solve fundamental questions of AR models. It is entirely possible that an AR-based model might in the future re-establish the SOTA. The fact that VAR has been selected for the Best Paper Award in NeurIPS 2025 is a testament to its potential to have long-term impact in image/video generation.
>
> Since both VAR and Switti are one of the most popular choices for AR-based models to achieve SOTA in AR image generation, successfully applying IGG on these models (which we did) thus ensures that it can be adapted to any modern AR-based model that deploys a similar scale-wise prediction strategy.
>
> # [W3] Concern regarding ablation studies
>
> In section 6.4 and 6.5 of the main paper, we have already conducted extensive analysis on the models’ behaviours and concluded at least 4 insights as below:
> - The metric analysis (Section 6.4) explores hyperparameter sensitivity, revealing that optimal FID coincides with an equilibrium between evenness and divergence.
> - The first analysis in Section 6.5 shows that an equilibrium between evenness and divergence is what drives FID optimality, not either metric in isolation.
> - The fixed schedule ablation analyse in Section 6.5 confirms that the ratio-based schedule is structurally necessary, as applying uniform guidance across scales contradicts the coarse-to-fine nature of scale-wise AR generation.
> - The sliding-window analysis in Section 6.5 reveals that the localising attention to the neighbourhood of a token is sufficient in determining its importance.

---

### Official Review · Reviewer_o9zx · 2026-03-16

**Soundness:** 2
**Presentation:** 2
**Significance:** 2
**Originality:** 2
**Overall Recommendation:** 4
**Confidence:** 3

**Summary:**

This paper IGG discovered the flaw of CFG design in the current SwAR models during class conditioned sampling, pointing out the mismatch in the guidance and the generated tokens, and based on that proposed a information aware guidance by using different weight for different tokens during sampling. Theoretical analysis and empirical results are shown in the paper.

**Compliance With Llm Reviewing Policy:**

Affirmed.

**Final Justification:**

I have raised the score because the rebuttal well addressed my concerns.

**Key Questions For Authors:**

+ In Figure1, how is the heatmap for weak/strong guidance calculated? If that is the attention score to the semantic token, your method is bound to have better heatmap because you are consistently increasing the weight of some the tokens to the guidance, and it will result in certain regions have "stronger" guidance. Is it the case?
+ In section 4, you have hypothesised that "If the guidance mechanism is agnostic to semantic importance, the divergence between p and q should remain minimal." However, it should be reasonable for the distribution for foreground and background to diverge? Have you evaluated the score unconditional cases to alleviate the inherent distribution divergence between foreground and background tokens?

**Limitations:**

It might be good to further evaluate the soundness of the hypothesis and show qualitative improvements over the previous guidance method CFG.

**Strengths And Weaknesses:**

**Strengths**
+ The insight of analyzing the potential mismatch between the guidance and token generation for SwAR is inspiring, which shows the potential to further improve the current guidance techniques.
+ The proposed IGG is efficient, which brings minimal computational cost to the original CFG.
+ Extensive experiments have been done to show the improvement on the quantitative metrics.

**Weaknesses**
+ The results shown in Figure1 and Figure3 cannot show the superiority of IGG over CFG: only the cactus example shows a failure for the CFG.
+ The conclusion in Section 4 is not derived in a very rigorous way: an ablation on the foreground / background distribution divergence under unconditional sampling should also be studied as a baseline.

---

> ### Author Rebuttal · Authors · 2026-03-31
>
> ## [W1.] Explaination about the figures' examples
>
> IGG helps improve the quality of generated samples, as shown in different examples in the paper. For example, in Figure 1, we observe that, by receiving more guidance signals, the chicken in the IGG picture forms its face and body texture sooner, resulting in higher quality and fewer artefacts. The chicken image in the CFG case, on the other hand, suffers from black-stripe artefacts from the guidance signal. Similar to the above case, the examples in Figure 3 show a net improvement in prompt coherence. In the CFG case, due to the weaker signal in the complex regions, one of the four examples contains an error for each prompt: the horse rider got blended to the horse, or the cloud dragon is missing its arms.
>
> ## [W2.] Supporting evidence for Section 4's hypothesis.
>
> According to the request of the reviewer, we have performed unconditional sampling of 1,000 images and computed the average evenness and divergence scores. As expected, the yielded scores were 0.999 and 0.044 respectively, significantly worse than conditional sampling (even for vanilla CFG, which yielded 0.741 and 0.623). Illustrative samples are available here: https://anonymous.4open.science/r/igg-6356/var_uncond.png. This result confirms that the observed divergence between the foreground and background distribution is caused by applying guidance.
>
> ## [Q1.] How is the heatmap for weak/strong guidance calculated?
>
> The heatmap in Figure 1 (and also Figure 2) is not calculated by the attention score. Instead, it depicts the guidance “nudge” received by each token, that is, the difference between the associated conditioned and unconditioned predictions $p_\theta(s_k|c) - p_\theta(s_k)$. As revealed in Section 4, guidance in diffusion models tends to prioritise semantically important tokens while scale-wise AR models lack this mechanism. IGG addresses this shortcoming by concentrating guidance towards foreground regions, achieving this through the attention mechanism. The heatmap in Figure 1 visualises the difference in guidance (compared to the vanilla CFG) that results from this modification.
>
> ## [Q2.] Question about divergence between foreground/background
>
> As mentioned above, we have investigated the foreground and background distributions under unconditional sampling and found that there is negligible divergence (0.044) between these distributions. This result supports our hypothesis, consequently confirming that the observed divergence is driven by the guidance process.

---

> > ### Author Rebuttal · Reviewer_o9zx · 2026-04-06
> >
> > Thank the authors for the explanation, they well addressed my concerns about the quality improvement. I will raise my score towards a positive one.

---

### Decision · Program_Chairs · 2026-04-30

**Decision:**

Accept (regular)

**Comment:**

This paper proposes a CFG method for scale aware autoregressive models for image generation. The key insight is that it would be beneficial to put different guidance weights on tokens based on their semantic importance. Experiments are conducted on both class and text conditioned image generation which demonstrates positive gains from the proposed method.

All reviewers acknowledge the soundness of the proposed method, though there are also minor critiques on the heuristic nature of it. Reviewers also find the experiments convincing in general, but with more room for improvement such as experimenting with stronger baseline models. The authors' rebuttal effectively addressed large part of the concerns, which results in increases of ratings. The AC believes that this work provides value to the AR generative modeling community and recommend accepting.